# Preparation and Formula Analysis of Anti-Biofouling Titania–Polyurea Spray Coating with Nano/Micro-Structure

**Yuanzhe Li** [1] , **Boyang Luo** [1] , **Claude Guet** [1] , **Srikanth Narasimalu** [2] **and Zhili Dong** [1,*]

[1]    School of Materials Science & Engineering, Nanyang Technological University, Singapore 639798, Singapore
[2]    Energy Research Institute @ NTU (ERI@N), CleanTech One, Singapore 637141, Singapore
*    Correspondence: zldong@ntu.edu.sg; Tel.: +65-6790-6727

**Abstract:** This paper proposes the preparation and formula analysis of anti-biofouling Titania–polyurea ($TiO_2$–SPUA) spray coating, which uses nano-scale antibacterial and photocatalytic agents, titanium dioxide, to construct regularly hydrophobic surface texture on the polyurea coating system. Through formulating analysis of anti-biofouling performance, it is found the causal factors include antibacterial $TiO_2$, surface wettability and morphology in order of their importance. The most optimized formula group is able to obtain uniform surface textures, high contact angle (91.5°), low surface energy (32.5 mJ/m$^2$), and strong hardness (74 A). Moreover, this newly fabricated coating can effectively prevent Pseudomonas aeruginosa and biofilm from enriching on the surface, and there is no toxins release from the coating itself, which makes it eco-friendly, even after long-time exposure. These studies provide insights to the relative importance of physiochemical properties of Titania–polyurea spray coatings for further use in marine, as well as bio medical engineering.

**Keywords:** Titania–polyurea ($TiO_2$–SPUA) spray coating; morphology study; surface wettability; anti-biofouling study; surface hardness; formula analysis

---

## 1. Introduction

Photocatalytic titanium dioxide ($TiO_2$) is a semi-conductive material, which could produce free hydroxyl and oxygen species (reactive oxygen species) with strong oxidation-reduction ability by photosynthesis reactions [1]. Organic compounds and inorganic oxides could be decomposed under illumination [2]. The photocatalytic character of $TiO_2$ has successfully made it widely applied in various types of bacteria and biofilms contamination abatement [3].

The conditioning of photoinduced bactericidal activity on $TiO_2$-films surface experiments by Pleskova et al., also states that reactive oxygen species (ROS) is the key to causing destruction of bacteria [4], and its effectiveness of bactericidal activity of $TiO_2$-films depends on several factors, such as time of UV irradiation and thermal and chemical treatment of films on the bactericidal activity [5]. Recently, Kim et al., reported the reaction of titanium dioxide on Streptococcus mutans biofilm, confirming that the antibacterial effect could be indicated by causing the photocatalytic reaction of $TiO_2$ in S. mutans biofilm, even at the wavelength of visible light (405 nm) [6]. Besides, titanium dioxide has extremely strong characters of sterilization, deodorant, mould proof, and antifouling function of self-cleaning [7].

Polyurea coatings with better performance than traditional polyurethane and extreme properties, such as rapid cure, insensitivity to humidity, flexibility, high hardness, tear and tensile strength [8], and chemical and water resistance provide a great platform for the fabrication of new type of composite coatings [9]. With proper primer and surface treatment, excellent adhesion to steel and other substrate

materials could be achieved [10]. In addition to advantages of polyurea over other coatings, the titanium dioxide could form nano-micro scale regular textures on the coating surface, which might contribute to the hydrophobic and drag reduction performance and also the control of bio-adhesion [11]. Moreover, the combination of these two materials, which is designed for the prevention of biological attachment, could maintain the immersion structure, such as ships as well as the cleanness of sonar in marine, is seldom documented.

Typically, the hull antifouling paint is made of paint, toxic pigments, solvents, additives, and other components [12]. One state-of-the-art antifouling paint that is used in ships in seawater as well as fresh water reviewed by Lin et al., is indicated in Figure 1a. When the antifouling paint confronts with water, the toxic pigment cuprous oxide releases, leaving only the adhesive shell afterwards [13]. However, after a long enough period of exposure, the adhesive shell becomes so thick that the rate of toxins that release into the meager water layer is below the critical value for bactericidal effect and forms a thick, sealed, "sandwich" structure with thickness of 1000 to 1200 μm. Such a "sandwich" will produce a lot of interfacial stress, and peel from the substrate, resulting in very rough surface morphology [12,14]. Additionally, the toxin, cuprous oxide, also imposes the risk to the environment [15].

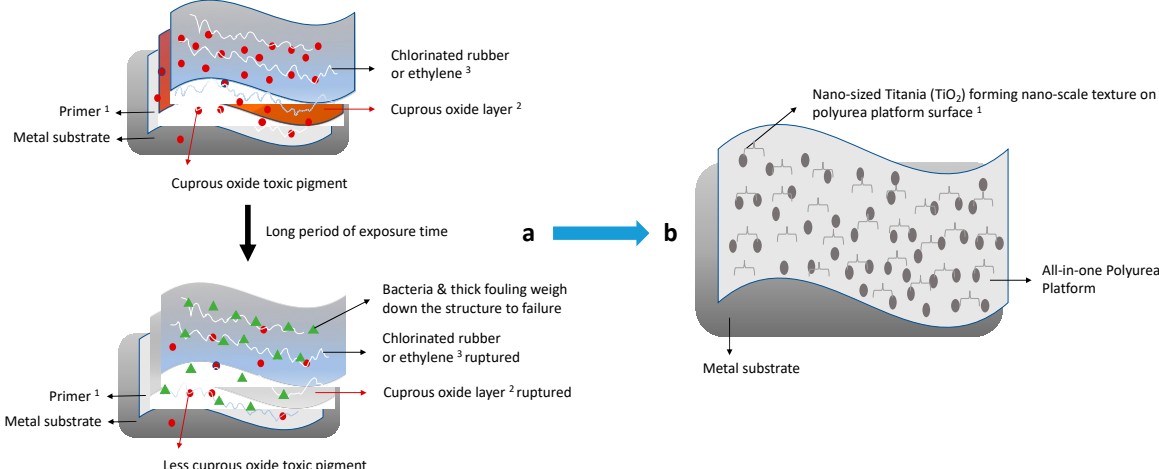

**Figure 1.** Schematic diagram of typically antifouling paint and all-in-one Titania–polyurea spray coating. (**a**) typically antifouling paint and ruptured structure after long-time exposure, and (**b**) all-in-one anti-biofouling Titania–Polyurea spray coating (TiO$_2$–SPUA).

In this paper, the preparation and formula analysis of such coatings according to their chemical composition, surface morphology, and wettability, which contributes to different bioactive properties, are discussed. Different from high consumption of nano-titanium dioxide and complicated forming process for better antibiofouling performance in most researches [16,17], this Titania–polyurea (TiO$_2$–SPUA) spray coating greatly reduces the use of nano-TiO$_2$ wt.% by forming all-in-one design coating system easily, as shown in Figure 1b. It is believed that the coating that we aimed at and developed can greatly slow down the biofouling process for marine structures, increase the efficiency of the ecosystem with less energy consumption, and extend the maintenance cycle.

Furthermore, this newly fabricated TiO$_2$–SPUA coating system, especially in its microstructure properties and formulating analysis of anti-biofouling performances, have not been recorded yet.

## 2. Materials and Methods

### 2.1. Materials

50–80 wt.% isocyanates (TDI)–component "A" and 50–90 wt.% polyether amines (Polyoxypropylenediamine)–component "B" were provided by Dragonsheild-BC$^{TM}$, Specialty Products Inc. (Washington, DC, USA). Versalink$^®$ P-1000, and it was alternative choice of Component "B"

purchased from Versalink (Essen, Germany), Air Products and Chemicals Inc. (Allentown, PA, USA). Low-surface-energy deformer, polydimethylsiloxane (PDMS), and isopropyl alcohol (IPA), from Sigma Aldrich (Bangkok, Thailand) were used as provided. The anatases nano-titanium dioxide with mean size of 39.6 nm and 0.293 polydispersity was fabricated by the sol-gel methods [2].

*2.2. Experimental Procedure*

2.2.1. Formulations and Fabrication of Titania–Polyurea (TiO$_2$–SPUA) Spray Coating

Table 1 indicated the three typical group formulations of TiO$_2$–SPUA coatings. The first two groups (G1 & G2) of TiO$_2$–SPUA coating formulations took consideration of the excess isocyanate compensates for the 'loss' of isocyanate-groups during storage due to humidity and/or application, as well as the 'losses' from additives in Part B [8], and the increased weight percentage of Part A slightly higher than Part B with index range 1.66 in Group 1 (G1) and relatively low index range 1.50 in Group 2 (G2) [10]. The Group 3 (G3) of TiO$_2$–SPUA coating prescription remained the index range 1.00, the only difference between T-PG3E1 (air-drying) and T-PG3E2 (oven-drying) is drying methods, which might contribute to the different configuration of the surface [17], as shown in the pre-experiment.

**Table 1.** Table of TiO$_2$–SPUA coating formulations (Group 1–3).

| Formulation Code Name | Part A | | Part B | | | |
| --- | --- | --- | --- | --- | --- | --- |
| | Component "A" | Component "B" | P1000 | TiO$_2$ | PDMS | IPA |
| PG1E1 | 62.5 | 15.0 | 17.5 | – | – | – |
| T-PG1E2 | 62.5 | 14.8 | 22.3 | 0.4 | – | – |
| T-PG1E3 | 62.5 | 14.7 | 22.0 | 0.4 | 0.4 | – |
| T-PG1E4 | 62.5 | 14.7 | 22.0 | 0.4 | – | 0.4 |
| PG2E1 | 60.0 | – | 40.0 | – | – | – |
| PG2E2 | 60.0 | 8.0 | 32.0 | – | – | – |
| T-PG3E1 | 50.0 | 10.5 | 36.5 | 1.5 | 1.5 | – |
| T-PG3E2 | 50.0 | 10.5 | 36.5 | 1.5 | 1.5 | – |

Note: T is for formulation obtained TiO$_2$, P is short for "Polyurea"; G is short for "Group" with same wt.% of Part A/Component "A"; E is short for "Experiment" with different formulation of Part B; e.g., T-PG1E2 means TiO$_2$–polyurea Group 1 (62.5 wt.% Component "A") Experiment 2.

For a better mixture of components, both of the chemicals were raised to 70 °C to decrease their viscosity [18]. Subsequently, Part A and Part B were put into Kakuhunter SK-300TVSII mixer (Shashin Kagaku Pte Ltd., Kyoto, Japan) for 180 s with vacuum level of 0.5 kPa to remove bubbles. The revolution speed and rotation speed were set at 580 rpm and 1700 rpm, respectively [10]. After mixing, the TiO$_2$–SPUA coating was sprayed with 2.0 mm thickness into the Teflon mould with dimension of $100 \times 100 \times 10$ mm$^3$. Finally, all of the samples were put into oven under 70 °C (except for T-PG3E1) for curing of 48 h.

2.2.2. Surface Characterizations

The surface morphology and chemical composition of TiO$_2$–SPUA coating were characterized by scanning electron microscopy (Hitachi S-4700, California, CA, USA) [17] that was attached with a Bruker AXS Quantax 4010 energy dispersive X-ray spectrometer (EDX, Karlsruhe, Germany) [19]. OCA15 plus (Dataphysics, Filderstadt, Germany) was employed to measure the contact angles while using distilled water. Surface energy was measured using three different liquids, diiodomethane, ethylene glycol, and distilled water, and SCA 20 software (Dataphysics) was used to analyze data from the regression line in a suitable plot [20,21].

### 2.2.3. Preparation of Bio-Medium and Bioassays

All of the tests were conducted while using a nutrient-rich artificial fresh water medium referred to Yuan et al. [22]. A well-studied aerobic Pseudomonas sp. NCIMB 2021 bacterium was obtained from the National Collection of Marine Bacteria (Sussex, UK). CDC biofilm reactor (BioSurface Technologies Corp., Bozeman, MT, USA) was used to grow Pseudomonas enriched cultures from Sentosa (SG) seawater under different shear force and continuous flow conditions at 30.0 °C [7,22]. The source of UV-irradiation was a Philip TUV15W lamp (Eindhoven, Netherlands) with power densities at 32.12 mW/cm$^2$, i.e., the maximum intensity was at 353 nm, which is under the UV-A region (315–400 nm). The distance between the light and specimens was set to 7.0 cm, and the interval for each UV treatment was 8 h/day to simulate the practical sunlight radiation. Specimens that were cut from coating samples were served as removable coupons and ASTM method and E2562-12 Manual was followed. Live/Dead BacLight$^{TM}$ Bacterial Viability Kit for microscopy (Thermo Fisher Scientific, Waltham, MA, USA) was employed to undertake the detection of biofilms [23]. Pyruvate and organic carbon were inserted into the carboy lid to speed up the enriched culture in CDC biofilm reactor [24–26]. CVD titanium dioxide coated on AlZr (5.25% Zr) 2 µm 316 L polished substrate [27] and fluoro-modified elastomeric polyurethane, followed by previous drag-reduction work [11], were also involved as two control groups for the analysis of biofilm-attachment mechanism and formula optimization.

### 2.2.4. Surface Hardness Test

ASTM-D-2240-00 Type A Teclock durometer was used to test the hardness of TiO$_2$–SPUA coating and its impact resistance [17,21].

## 3. Result and Discussion

### 3.1. Formulating of Surface Features

The prescription added more Part A than Part B, which was considered the possibility of self-healing and might have un-reacted isocyanates inside the fracture surface, to have continuous reaction with water as per the following chemical reaction [10,28]:

$$R\text{-}NCO + H_2O \rightarrow R\text{-}NH_2 + CO_2 \tag{1}$$

This off-white colored Titania–polyurea (TiO$_2$–SPUA) spray coating, as shown in Figure 2, successfully inherited the fast reactivity of the polyurea. The curing time of all polyurea spray coating was within one hour. In the meantime, vacuumed mixer obtained by the spraying equipment could contribute to realize suitable mixing and remove the waste product and carbon dioxide shortly [29]. Besides, different drying methods, air-drying and oven-drying, were also tried under Group 3, would have an impact on the curing time of TiO$_2$–SPUA coating surface in the mould [30]. As for foam inside such a coating, both technology-based methods (e.g., vacuum degree inside the spraying equipment) and chemical-based prescription (e.g., defoamer, such as PDMS and IPA) have been conducted to address this issue [10]. To conclude, the ideal solution for the foam issue should be preventing the CO$_2$ generation and diffusion from the reaction between moisture in the air [31]. The shorter drying time TiO$_2$–SPUA coating needs, the smaller amount of foam caused by carbon dioxide that it would generate.

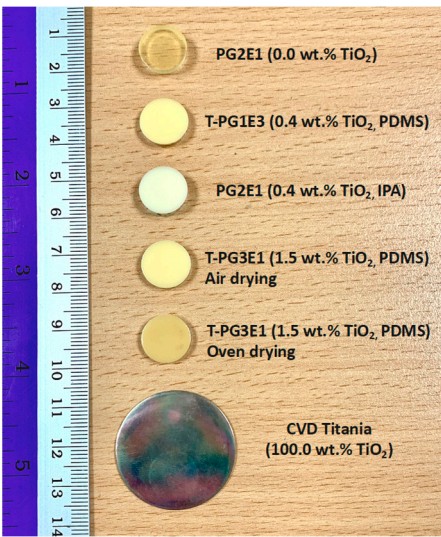

**Figure 2.** Photographs of Titania–polyurea spray coating samples.

### 3.1.1. Morphological and Chemical Composition Analysis

For a polyurea coating group without titanium dioxide, the morphological image was relatively flat when compared with $TiO_2$–SPUA coatings, even other liquid additives, such as PDMS and IPA solution, would not cause any modification of the coating surface structure [17,32]. While, for the SEM images shown in Figure 3a–c, only the increment in nano-titanium dioxide could directly lead to the surface textures change, and these surface structures might have a good orientation of the water flow and detachment of biofilm [33], which was caused by the spraying and curing during preparation. Additionally, it was gratifying to observe that the nano-$TiO_2$ particles did not generate agglomeration and had a favorable dispersion in polyurea coatings. As the nano-titanium dioxide weight percentage went higher, from 0.4 wt.% $TiO_2$ shown in Figure 3b to 1.5 wt.% $TiO_2$ shown in Figure 3c, textures that formed by nano-titanium dioxide would become more distinct.

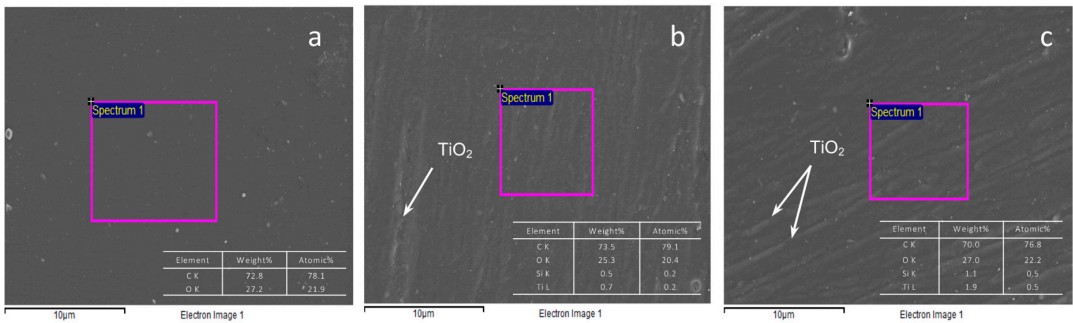

**Figure 3.** Scanning Electron Micrographs showing surface morphologies of sample (**a**) PG2E2, (**b**) T-PG1E2, and (**c**) T-PG3E2.

It could be summarized that, as the nano-$TiO_2$ weight percentage continuously going high, the surface morphology would cohere hydrophobic and homogeneous surfaces along with drag reduction character at first. It is important to add that, as the weight percentage continuously increased, the hydrophilic character from nano-$TiO_2$ would gradually take the lead, and its internal agglomerating force would end up with rough coating surface morphology as compared to the original smooth finish. Moreover, the profile degree, force orientation, and wettability of $TiO_2$–SPUA coatings would also be affected [34]. From the investigations of contact angle (CA) and surface energy (SFE) analysis subsection, these inferences were also confirmed. From the imaging of EDX spectra, the Ti and Si

signal appeared, and the signal intensity would continuously increase along with the increment in nano-TiO$_2$ and defoamer PDMS.

### 3.1.2. Surface Wettability

Pure titanium, as well as titanium dioxide surfaces, performed as hydrophilic. However, from the overall surface energy (SFE) and contact angle (CA) result in Table 2, the hydrophobic character behavior of TiO$_2$–SPUA coating was exactly different. The contact angle changed by the increase of TiO$_2$ wt.% was also indicated, respectively, in Figure 4a–c. It could be not only due to its small weight percentage and its minor texture structure on the coating surface, but its dispersion inside the polyurea base coatings and use of low surface energy defoaming agent could also influence the polar component, CA, SFE, and even biofilm test result [34].

**Table 2.** Values (mean ± standard deviation) of contact angle (CA) and surface energy (SFE).

| Formulations & Effect | PG1E1 | T-PG1E2 | T-PG1E3 | T-PG1E4 | PG2E1 | PG2E2 | T-PG3E1 | T-PG3E2 |
|---|---|---|---|---|---|---|---|---|
| TiO$_2$ (wt.%) | – | 0.4 | – | 0.4 | – | – | 1.5 | 1.5 |
| PDMS (wt.%) | – | – | 0.4 | – | – | – | 1.5 | 1.5 |
| CA (°) | 61.5 ± 2.7 | 66.9 ± 2.9 | 73.0 ± 4.9 | 64.4 ± 1.4 | 68.4 ± 6.9 | 63.8 ± 1.9 | 88.5 ± 6.4 | 91.5 ± 2.6 |
| SFE (mJ/m$^2$) | 57.3 | 49.4 | 45.8 | 51.5 | 47.1 | 52.3 | 37.2 | 32.5 |

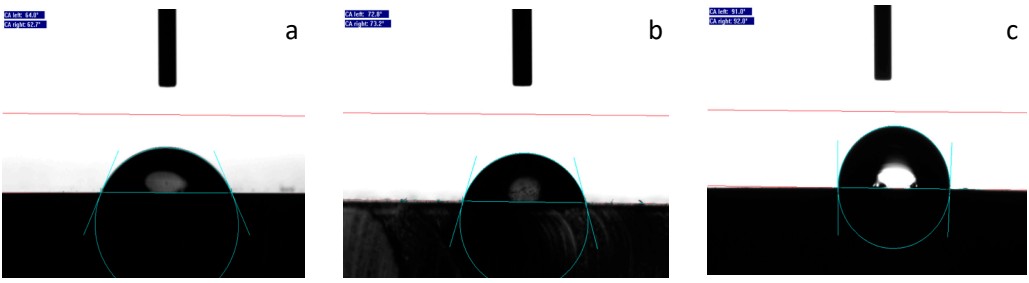

**Figure 4.** Surface contact angle (CA) of coating surface (**a**) contact angle of PG2E2 (0.0 wt.%TiO$_2$) 63.8° ± 1.9°, (**b**) contact angle of T-PG1E2 (0.4 wt.%TiO$_2$) 66.9° ± 2.9°, and (**c**) contact angle of T-PG3E2 (1.5 wt.%TiO$_2$) 91.5° ± 2.6°.

These CA and SFE results should also be related to morphology. It is worth noting that increment scenarios of CA would only happen at small weight percentage scale of nano titanium dioxide. As the increase of nano-titanium dioxide particles wt.%, the nano-texture and low surface energy structure would gradually disappear. High weight percentage of hydrophilic nano-titanium dioxide at 5% and 10% would lead to the decrement of CA to 68.0° and SFE would raise as high as 50.3 mJ/m$^2$.

Otherwise, based on the fact that low surface energy chemistry and nano-textured morphology of the hydrophobic coating could result in reduced protein adsorption and bacterial attachment [35], it was not found that there was low attachment of bacteria and biofilms in the following biofouling assays. Additionally, the hydrophobicity of such coating surface might reduce the flow resistance and also offer a new research direction for drag reduction.

### 3.2. Formulating of Anti-Biofouling Performance

Cells with compromised membranes that are considered to be dead or dying will stain brightly red, whereas the cells with an intact membrane will stain brightly green. The dark spot and slightly blur background were due to the artifacts of substrate structures, and even surface reflection/refraction under the Bacterial Viability Kit, respectively, rather than biofilms. In addition, more quantified information of Pseudomonas aeruginosa in this pilot bio-assays is provided in Table S1.

### 3.2.1. Nano-Titanium Dioxide vs. Photocatalytic and/or EPS Degradation & Morphology

During the early stage of the Pseudomonas aeruginosa attached, reversible attachment is secreted by bacteria with tightly-bound EPS (TB-EPS), which keeps cells together in clusters and loosely-bound EPS (LB-EPS) bonding different bacteria clusters together to form stable micro-colonies [6]. The continuous production of EPS by the bacteria community (made up of protein, polysaccharides, eDNA, bacterial lytic products, and compounds from the host) provides the biofilm structural integrity [36,37].

- High weight percentage of nano-titanium dioxide ($TiO_2$ wt.%) in the coating system may cause the photocatalytic degradation and EPS degradation to inhibit the reversible attachment of biofilm.

The strategy to tackle Pseudomonas for high weight percentage of nano-titanium dioxide ($TiO_2$ wt.%) is the photodegradation of the EPS matrix [38]. Through the observation of the live cells while using the Bacterial Viability Kit, there was no sign of any live/dead Pseudomonas or any ruptured structure of microbial cells under higher weight percentage of the nano-$TiO_2$, i.e., there was no indication of attached biofilm or any other microbial cells on the coating surface, except for the artificial defects (dark areas) that are caused by long term exposure and etching from the surface as shown in Table 3.

**Table 3.** Live/dead microscopy of control group with pure nano-scale $TiO_2$.

| Formulation Code Name | Surface Features | Nano-$TiO_2$ wt.% | 60 ± 5 rpm/10 days (Low Shear Force) | 240 ± 5 rpm/10 days (High Shear Force) |
|---|---|---|---|---|
| CVD $TiO_2$ Surface (Control Group 1)-Live | Super hydrophilic (CA < 5°) | 100.0 | | |
| CVD $TiO_2$ Surface (Control Group 1)-Dead | Super hydrophilic (CA < 5°) | 100.0 | | |

Under high shear force, the etching of CVD titanium dioxide surface was even more serious than the ones under low shear force, as indicated in Table 3. Through characterized by 90 plus nanoparticle size analyzer (Brookhaven, New York, NY, USA) for a leaching test, nano-titanium dioxide leaking from CVD titanium dioxide surface was observed at the region between 35.0 and 50.0 nm in the seawater medium after 10 days. Hence, this nano-titanium dioxide from etching in the CDC system was supposed to work on the photodegradation of Pseudomonas [39]. The schematic diagram in Figure 5a demonstrated the detailed processing of how nano-$TiO_2$ particles produce reactive oxygen species (ROS) [40], inhibit DNA form replication, photodegrade EPS and many other proteins, damage cell membrane, and even interrupt the electron transportation of cells [41].

- Low weight percentage of nano-titanium dioxide ($TiO_2$ wt.%) in the coating system may use photocatalytic degradation to inhibit the attachment of biofilm by damaging microbial membrane and quorum sensing.

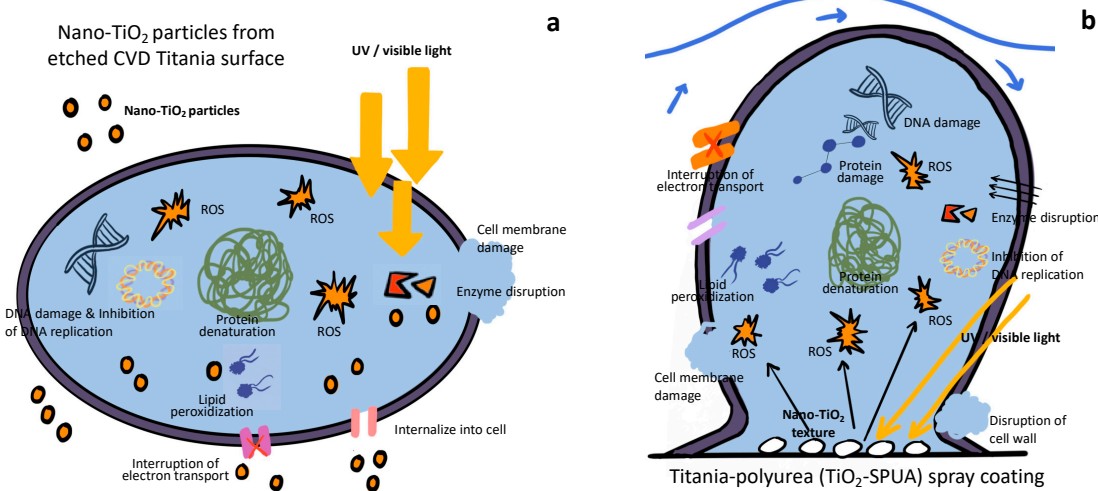

**Figure 5.** Schematic diagram of different anti-biofouling processing caused by nano-TiO$_2$ photodegradation (**a**) nano-TiO$_2$ particles from etched CVD TiO$_2$ surface, and (**b**) Titania–polyurea (TiO$_2$–SPUA) spray coating.

At these lower formulation groups of 0.4 wt.%TiO$_2$ (T-PG1E2), 1.5 wt.%TiO$_2$ (T-PG3E2), and even 5.0% wt.%TiO$_2$, with the decrement of weight percentage of nano-titanium dioxide (TiO$_2$ wt.%), the Pseudomonas inhibition and photodegradation rate would become lower than the one with higher TiO$_2$ wt.% (CVD TiO$_2$ Surface). The first-stage EPS would be already stabilized, and biofilm might also reach the irreversible attachment [37,42]. However, the good news was that, through leaching test by nanoparticle size analyzer, only fewer nano-particles could be detected than CVD ones. The result also indicated its potential application for long term exposure in the biofouling environment.

A broken structure of microbial cells and biofilm (DNA of Pseudomonas), which were stained red, could be clearly observed. The internal substances and degraded components [43] from microbial cells are indicated in the Table 4. Though Pseudomonas structure was still destabilized from the photocatalytic degradation and reactive oxygen species (ROS) from the nano-TiO$_2$, as indicated in Figure 5b, the biofilm was already mature when comparing with the high TiO$_2$ wt.% groups. Additionally, the free adhesion of biofilm would possibly result from the communication of the Pseudomonas clusters, which is so called quorum sensing/quenching [44].

**Table 4.** Live/dead microscopy of control group indicating the effect from nano-TiO$_2$ wt.%.

| Formulation Code Name | Surface Features | Nano-TiO$_2$ wt.% | 60 ± 5 rpm/10 days (Low Shear Force) | 240 ± 5 rpm/10 days (High Shear Force) |
|---|---|---|---|---|
| T-PG1E2 | Hydrophilic (5° < CA < 90°) | 0.4 | | |
| T-PG1E2 | Hydrophilic (5° < CA < 90°) | 0.4 | | |

**Table 4.** *Cont.*

| Formulation Code Name | Surface Features | Nano-TiO$_2$ wt.% | 60 ± 5 rpm/10 days (Low Shear Force) | 240 ± 5 rpm/10 days (High Shear Force) |
|---|---|---|---|---|
| T-PG3E2 | Hydrophobic (90° < CA < 150°) | 1.5 |  |  |
| T-PG3E2 | Hydrophobic (90° < CA < 150°) | 1.5 |  |  |

Moreover, the coating samples with regular roughness and morphology would also be conducive to the detachment of biofilm [45] by changing the contact area and air-water interface [46] between the cells and coating surface, as demonstrated in Figure S1a. For nano/micro-scale surface textures (<10 μm), which is smaller than the cell size of Pseudomonas or its cluster (>20 μm), the liquid mobile phase biofilm could shear freely at the air-water interface with small resistance, which resulted in wall slippage [47] and lower friction coefficient and flow resistance. Such a slip effect indicated in Figure S1b would also contribute to the drag reduction effect of microfluid, as proven in previous research [11,48].

At present, there are few studies on substrate interface with certain morphological characteristics, especially those on nanometer or micron level morphological characteristics and substrate interface film covering. Machado et al. [49] used chemical etching techniques to study the surface energy of nano-scale PVC materials and their effects on initial cell adhesion. It was found that the surface energy of the material and the initial adhesion of cells to the interface would be changed due to the change of roughness and morphology and the ability to attach was reduced. Furthermore, since the cell sizes are different, there is no uniform standard for different bacteria [50], and the relationship between roughness and morphology and cell adhesion remains to be further studied.

3.2.2. Hydrophobicity/Hydrophilicity vs. Biofilm Adhesion

- The hydrophobic surface may reduce the adhesion of the biofilm more significantly than hydrophilic ones under high shear force.

For most of the hydrophobic surfaces, the electrostatic interaction and cohesive strength between the biofilm and the surface are weaker than the hydrophilic [51], so the biofilm on the hydrophobic/superhydrophobic surface can be even easier to fail and lead to the detachment event than the hydrophilic one [52]. PDMS, as a polymeric matrix of silicone coatings, are also well known for their smoother and non-stick surfaces relative to other polymeric matrices. Likewise, through vertical comparison between super hydrophobic/hydrophilic surface without effect of any nano-titanium dioxide in the Table 5, it could be found the attachment of biofilm (Pseudomonas) at the superhydrophobic surface (fluoro-modified elastomeric polyurethane) was relatively lower than the super hydrophilic (concrete coupon) ones.

**Table 5.** Live/dead microscopy of control group excluding the effect from nano-TiO$_2$.

| Formulation Code Name | Surface Features | Nano-TiO$_2$ wt.% | 60 ± 5 rpm/10 days (Low Shear Force) | 240 ± 5 rpm/10 days (High Shear Force) |
|---|---|---|---|---|
| Concrete coupon of CDC biofilm reactor-Live | Super hydrophilic (CA < 5°) | 0.0 |  | |
| Concrete coupon of CDC biofilm reactor-Dead | Super hydrophilic (CA < 5°) | 0.0 |  | |
| Fluoro-modified elastomeric polyurethane (Control Group 2)-Live | Superhydrophobic (CA > 150°) | 0.0 |  | |
| Fluoro-modified elastomeric polyurethane (Control Group 2)-Dead | Superhydrophobic (CA > 150 °) | 0.0 |  | |

Additionally, it should be noted that biofilms on both of these two groups of polyurea coating samples were also shown to become micro-colony formation and biofilm maturation, which were definitely irreversible [53].

- Hydrophobic and hydrophilic coating surface may both potentially reduce the adhesion of the biofilm under high shear force by convection factors.

According to the horizontal comparison of each sample, the rotating speed (rpm) and shear force could result in difference adhesive image of live/dead Pseudomonas cells, which is mainly because of the convection around the biofilm. Under slow rotating speed (60 ± 5 rpm), the shear force was relatively low, which enable the rapid growth of the Pseudomonas, as fluid could contribute to the diffusive transport of the metabolic substrate and the surrounding environment. As the rotating speed went higher (240 ± 5 rpm), fluid would surround Pseudomonas cell clusters, but was still too weak to flush through them. Complex secondary flows were able to occur under this situation. Continuously increased rotating speed would gradually weaken the cohesive strength inside the biofilm cells and oscillating streamers would form on the downstream edge of a cell cluster [54]. Additionally, it is not hard to understand why under higher rotating speed (rpm), the attached biofilm for both coating samples were decreased more or less.

To conclude, setting all of the photodegradation factors aside, there is a preference between hydrophobic and hydrophilic coating surface for the attachment of the Pseudomonas. Although

the critical influence would be taken by the coating materials themselves, hydrophobic coatings will still show more significant reduction for the biofilm adhesion, especially at high shear force, than 5h3 hydrophilic ones [55]. Moreover, these reduction or inhibition of biofilm adhesion should be non-selective.

### 3.3. Formulating of Surface Hardness

As most of the coating surface might suffer different types of impacts from marine organisms as well as an undercurrent, an investigation of durability and impact resistance of $TiO_2$–SPUA coating was also very important [17,56]. Through supplementary optimizing design and numerical simulation of these newly fabricated coatings, rather than PDMS, deformer IPA did not exhibit significant increment of other mechanical performance, including surface hardness and even compression and tensile strength, which had no means to exhausted all.

The surface hardness results are shown in Table 6. Besides, the compression behaviour for these coatings, as shown in Figure S2, was also consistent with the hardness test results. It is not difficult to find that three main factors: 1) proportion of Component "B" (long chain) to P1000 (short chain) wt.%, 2) Deformer PDMS wt.%, and 3) $TiO_2$ wt.% would have the main influence for the performance of coating surface hardness. The increment of PDMS wt.% and P1000wt.% would decrease the surface hardness, while the surface hardness would significantly increase as $TiO_2$ wt.% and Component "B" wt.% grows. More information of the influence factors for these three main components to the surface hardness, including many other formulations, is also indicated in Figure S3 by regression lines.

**Table 6.** Values of surface hardness.

| Formulations & Effect | PG1E1 | T-PG1E2 | T-PG1E3 | T-PG1E4 | PG2E1 | PG2E2 | T-PG3E1 | T-PG3E2 |
|---|---|---|---|---|---|---|---|---|
| Component "B": P1000 (wt.%) | 1.5 | 1.5 | 1.5 | 1.5 | 0.0 | 4.0 | 3.5 | 3.5 |
| PDMS (wt.%) | 0.0 | 0.0 | 0.4 | 0.0 | 0.0 | 0.0 | 1.5 | 1.5 |
| $TiO_2$ (wt.%) | 0.0 | 0.4 | 0.0 | 0.4 | 0.0 | 0.0 | 1.5 | 1.5 |
| SH (A) | 77 | 98 | 48 | 75 | 50 | 92 | 72 | 74 |

To sum-up of the results that are indicated above, nano-scale $TiO_2$ and low-surface-energy PDMS were the most two critical factors for the optimization of the coating formulation. By using the matrix shown in Figure 6, performance that is derived from different weight percentage (wt.%) is indicated.

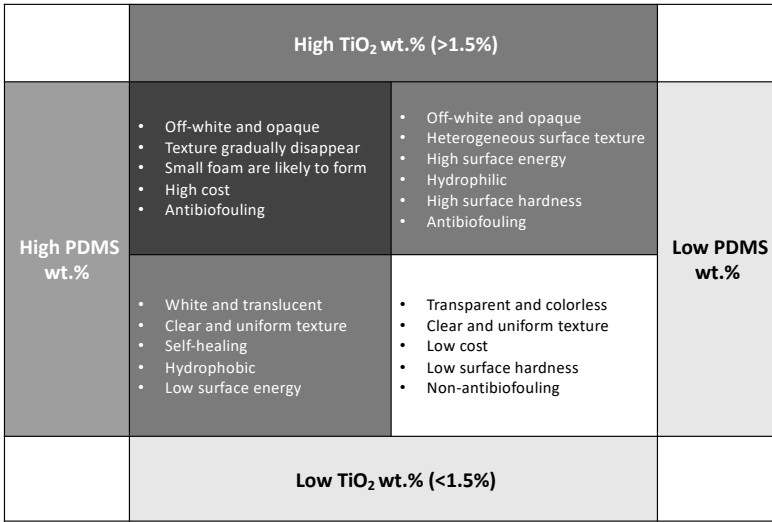

**Figure 6.** Different basic performance derived from coating formulation depending on weight percentage of $TiO_2$ and polydimethylsiloxane (PDMS).

Based on comprehensive consideration, the formulation group with high weight percentage (wt.%) of nano-titanium dioxide is not a good alternative to get high mechanical performance, as it might end up with relatively heterogenous texture and a poor dispersion of the nano-titanium dioxide [17]. On the contrary, the formulation group with lower nano $TiO_2$ wt.% and higher PDMS wt.% could not only prevent the biofouling attachment as previous group, but also obtain homogeneous texture, low surface energy, and many other good physicochemical performances. Additionally, these results will also become the good references for the practical applications and the optimization of the formulations.

For the formulation of such anti-biofilm and regular morphology $TiO_2$–SPUA coating fabrication, a Gaussian distribution should also be agreed with and the peak of its performance was supposed to be located at 1.0 to 5.0 wt.% of nano-titanium dioxide. One, in particular, is these biofilm attachment assays should be treated as pilot experiments at present. Further experiments are in progress to justify the relation among antibacterial character applied in practically marina environment.

This newly fabricated Titania–polyurea ($TiO_2$–SPUA) spray coating has the potential to replace the traditional antifouling paint, and its concept of all-in-one and rapid-cure polyurea platform would greatly shorten the painting and curing time and extend the maintenance intervals. There would be no more internal stress and peeling, resulting in very rough surface underwater, caused by the ruptured structure of the "sandwich" coating system after long time exposure. Moreover, there is no more toxins leaking from this Titania–polyurea ($TiO_2$–SPUA) spray coating, which is even more eco-friendly, and there is no worry about reaching the critical point for the antibacterial agents.

## 4. Conclusions

In this study, antibacterial Titania–polyurea ($TiO_2$–SPUA) spray coating with low consumption of $TiO_2$ (1.5 wt.% only) are fabricated. Regular surface textures and morphology, hydrophobic wettability, and low surface energy could be obtained during the process.

Through formulating analysis of free adhesion of Pseudomonas aeruginosa, it is found that the directly causal factors include: (1) Antibacterial $TiO_2$, (2) Surface wettability, and (3) Textures and morphology in order of their importance. Nano-titanium dioxide ($TiO_2$ wt.%) in the coating system may use photocatalytic degradation to inhibit the attachment of biofilm by damaging microbial membrane; hydrophobic wettability might reduce the adhesion of the biofilm more significantly than hydrophilic ones, especially under high shear force; and, nano-texture may also potentially reduce the biofouling adhesion. Moreover, the surface hardness is affected by: (1) the proportion of Component "B" (long chain) to P1000 (short chain) wt.%, (2) deformer PDMS wt.%, and (3) $TiO_2$ wt.%, which also has internal relationship.

The root cause of such features would be the two main additives, nano-titanium dioxide ($TiO_2$) and low surface energy defoamer (PDMS) inside the coating system. All of these studies would also provide good reference of formulation design and promising application for further application in marine, as well as bio-medical engineering.

**Supplementary Materials:** The following are available online at http://www.mdpi.com/2079-6412/9/9/560/s1, Figure S1: Schematic diagram of Pseudomonas detachment influenced by surface roughness and morphology (a) surface roughness/morphology and non-shear air-water interface, and (b) non-shear air-water interface and slip effect of TiO2-SPUA spray coating, Figure S2: Graph of Compression Stress vs Strain (a) Compression Stress vs Strain by PDMS wt.%, and (b) Compression Stress vs Strain by TiO2 wt.%, Figure S3: Influence factors for three main components to the surface hardness, Table S1: Bactericidal effects of different surfaces on Pseudomonas biofilms.

**Author Contributions:** Y.L. designed the experiments; Z.D. and S.N. contributed reagents and materials; Y.L. and B.L. conducted the formulation design, analysis, and fabrication; Y.L. wrote the paper; Z.D., S.N., and C.G. reviewed and provided corrections on the original draft.

**Funding:** This research was funded by MOE Academic Research Fund (AcRF) Tier 1 Grant Call (Call 1/2018)_MSE (EP Code EP5P, Project ID 122018-T1-001-077), Ministry of Education (MOE), Singapore.

**Conflicts of Interest:** The authors declare no conflict of interest.

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
