# Peer review of "Preparation and Formula Analysis of Anti-Biofouling Titania–Polyurea Spray Coating with Nano/Micro-Structure"

_coatings, doi:10.3390/coatings9090560_

Round 1

Reviewer 1 Report

General Comments

This paper approach an important pathway for scientific community and therefore congratulation for the researcher’s team. Actually, investigation may be rewrite much better. The paper is difficult to read, extensive and text may be synthesized. I suggest to the authors to higher manuscript review and clarify remarks before publish it. I hope that these suggestions may help to publish soon your paper.

On the other hand, the paper should be edited by a native-tongued English speaker as there are a number of grammatical issues with the manuscript.

Specific Comments

Abstract:

This section of paper will be widely read by scientific community. Therefore, this section should advance the main results.

Introduction:

- What is the scientific relevance of experimentation?

- Authors should include the main and specific goals of the investigation.

Materials and methods

Add a subsection called "Experimental procedure" where are defined the experimental phases and ordered the sequence of experimentation.

Results and discussion    

This part of paper is too much extensive and difficult to understand. The results must be written and discussed with the same arrangement to M&M.

Conclusions:

The authors do not clearly present their conclusions and more text should be.The main conclusions of the study may be presented in direct and short sentences

Reviewer 2 Report

The authors reported an interesting work in the field. However, the writing should be improved as further clarified, since there is a lack of an accurate discussion and important data that support the discussion of the results. Here are some comments that help improving paper quality and impact in the field:

Introduction

It should provide an accurate current state of the research field, but the present introduction shows some big issues. It is frequently found references which are inadequate or do not match with the statement provided by the text, and formatting of the mentioned references in the text body. For instance:

Lines 1-27, reference [1] – refers to air cleaning, it is not focused on photosynthesis reaction, it would be expected to include a more suitable reference mentioning this processor for a review paper.

Lines 28-29, reference [3] – does not address anti-biofilms, it studies the degradation of RhD B dye.

Line 31. This reference does not refer in the right format: “ Pleaskova, S.N. Et al., 2011” as others among the text.

Other issues:

Line 49: it is used “poison”!

Line 51-52: which reference by Olsen S. et al.? Reference [13] seems not to match with the text.

Line 57: again reference [12] seems not to match.

Line 63: reference [16] is not focused on anti-biofouling purpose it should be selected suitable references.

As a whole, a better state of the art should be provided regarding the immobilization of TiO2 in coatings.

Results and discussion

Lines 148-151: In Figure 3 or its caption should be clarified the TiO2 contents of the different samples.

Lines 152-159: This discussion should be in-line with the results shown not prior to those. Moreover later one, there is no linkage with results regarding the obtained CA and SFE and relation with these observations.

Lines 164-165: Is it really Pt in sample T-PG1E2?

Lines 179-188: These CA and SFE results should be related to morphology too.

Tables 4-6, from the provided photos it is not always clear the extension on biofilm formation on the different coatings, biofilm should be quantified. For instance for T-PG3E2 which is considered one of the best regarding anti-biofouling performance is seems to evidence higher extension of black spots, being unclear the smaller biofilm extension when compared with T-PG1E2.

Lines 240-264: A subsection is provided to discuss the relationship between morphology & roughness vs biofilm, but no data is provided regarding roughness on the prepared samples, and even reference [11] is not easily accessible. If a section is provided to discuss a parameter and relation with others, it should be provided experimental data to support it.

Lines 268-269: clarify the use of the word “hydrophilic”, it may be confused with hydrophobic.

Lines 298-316:

Why you are able to verify the decrease in hardness with PDMS wt.% increment? This could be better discussed.

The whole discussion should be improved and the Figure 7 doesn’t help too much, in fact, it is will confuse even more since the evaluated parameters behave differential and requirements for those are different at different applications, and what seems a good balance for one application may not be for another. But others issues were found, mostly related to the measure associated errors, which should be provided, for instance: i. what is the different between T-PG1E2 and T-PG1E4, their formulation seems similar and the SH(A) is quite apart, or between T-PG3E1 and T-PG2E3 which seems not to have a significant difference in their SH (A); ii. for the clear effect of component B, only PG1E1, PG2E1 and PG2E2 should be comparable. iii. for the TiO2 content effect the only comparable samples are the ones without TiO2 and the T-PG1E2 and T-PG1E4, since the others have others additives effects, and from T-PG1E2 and T-PG1E4 apparently no different in the formulations exists seems to be replicated.

An important missing coatings property assessment is the coating adhesion, which can be affected by the additives.

Lines 317-333: In my opinion, the index scores are not the most appropriated due to the same reason as prior mentioned, different experience conditions may require different parameters balance, and samples are apparently replicated and not independent samples.

Lines 334-335: there is no experimental data that supports the fewer toxins leaching. Leaching tests should be performed.

References format is not in accordance with the journal instructions.

Reviewer 3 Report

This paper describes a new coating. Due to the inclusion of TiO2, anti-biofouling is exerted by the effect of ROS and the addition of hydrophilicity.

The relationship between the surface morphology and the contact angle and the adhesion of the Biofilm has been described. However, it is not clear. The results for the Biofilm should be clearly presented.

The writing style needs to be improved. For example, 20nm should be 20 nm (need space). Table should not be involved in conclusion part. 

Round 2

Reviewer 2 Report

Dear authors,

I appreciated the improvements, but for my point of view, you need to provide some more data, even as support information to justify the results and discussion provided.

You should also improve the state-of-the-art. From this introduction, it is clear that you need to activate TiO2 with UV-Vis to become an efficient antimicrobial. Did you activate your coatings? If yes, how?

Sorry, but I cannot see any data on figure 3 or caption with the content of TiO2 in the different coatings. A Figure with its caption should provide as a whole the needed information to interpret it.

But Pt as also antimicrobial activity, this was the reason for my doubt. From the discussion, it seems you didn’t take into account the Pt present effect. I understood the colors, but looking only to the images, differences are not clear, and since in the section 2.2.3 you mentioned that you followed ASTM method and E2562-12 Manual, which is: Standard Test Method for quantification of Pseudomonas aeruginosa Biofilm Grown with high shear and continuous flow using CDS biofilm reactor, it is expected that you have those data. You should provide them in order to clarify the analysis. You can include them in support information.

I mean the roughness of the coatings themselves not with biofilm on it. You cannot discuss a parameter for your coatings without providing the data for those, reference 11 is not quite accessible. Coating roughness is a very important parameter for your coatings, mostly because you are introducing PDMS, which is a polymeric matrix of silicone coatings, which are well known of their smoother and non-stick surfaces relatively to others polymeric matrices. The same justify question 9. Which you have now improve it with lines 321-322 and again related with the PDMS smoother nature.

If you decide to not provide roughness data for your coatings, you should remove such a discussion section, you can just suggest some relation in a sentence or two.

You should look for table 7, you should provide the differences among the coatings in the table, improve the table legend: yes, but in table 7 this is not seen. Sorry, I mean T-PG3E1 and T-PG3E2, the same problem as above mentioned

Round 3

Reviewer 2 Report

Dear authors, you had improved the paper quality accordingly. 

All the best for the work.